# Sub-Picosecond Micromachining of Monocrystalline Silicon for Solar Cell Manufacturing

**Katarzyna Garasz \* and Marek Kocik**

Centre for Plasma and Laser Engineering, Institute of Fluid Flow Machinery, Polish Academy of Sciences, Fiszera 14, 80-231 Gdańsk, Poland; kocik@imp.gda.pl

**\*** Correspondence: kgarasz@imp.gda.pl; Tel.: +48-58-522-5117

**Abstract:** In this study a prototype sub-picosecond laser was investigated for cutting and scribing of silicon wafers. The Yb:KYW laser used for this investigation, unlike ultrashort systems used previously, generates pulses of 650 fs, i.e., between the pico and femtosecond range. The laser was placed in a micromachining setup, involving a galvo scanner and a telecentric lens. A study of the influence of the processing parameters on the crater width, depth, and quality of machining was carried out. The optimal parameters were found to be 343 nm, 200 kHz, 7 mm/s, and 15 pattern repetitions. The experiments were performed using samples of a silicon wafer of 210-μm thickness. The experimental results show that the sub-picosecond laser can be a promising and competitive tool for solar cell micromachining. In comparison to the commercially available ultrashort pulse laser systems, we find the sub-picosecond laser to be a more cost efficient and reliable source, than a femtosecond one. In addition, the prototype Yb:KYW design offers some unique parameters, such as repetition rate in the range of 100–400 kHz, UV wavelength or obtainable laser fluence close to the silicon ablation thresholds.

**Keywords:** sub-picosecond pulses; silicon wafers; photovoltaic cells; laser micromachining

## 1. Introduction

The rapid growth of the photovoltaics market requires advances in PV (photovoltaics) manufacturing and technologies that would translate into further cost reductions, as well as quality and efficiency increase of the solar cells [1]. Laser technology has already been widely applied in photovoltaic production for the last two decades, in many different ways (Figure 1) [2,3].

Among them, there are processes directly related to silicon wafers micromachining. Table 1 presents some examples of the femtosecond laser systems used for this purpose. They include dicing, nano and micro-structuration of the surface, and antireflection coating or nanoparticle absorbing [4–11]. According to Mazur et al., a simple irradiation of the silicon surface with ultrashort laser pulses may improve the cell efficiency [4–8]. One of the common micromachining processes engaged in increasing the efficiency of the solar cells is the so-called junction isolation (laser edge isolation). During the manufacturing process, a conduction path is created between the front and back surfaces of the wafer, all around its edge. In that direct process, the laser scribes a groove on the top surface of the silicon wafer, around the cell's periphery [2,12].

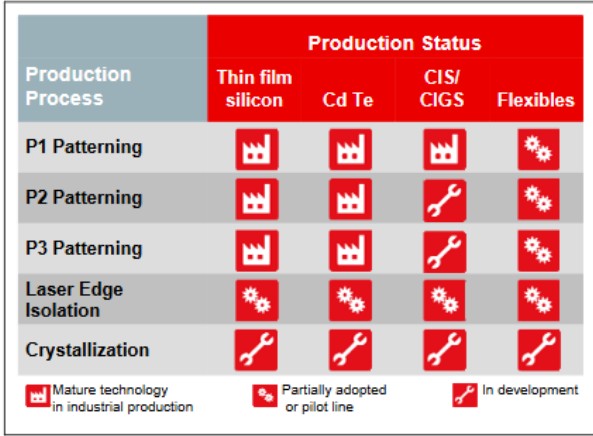

**Figure 1.** Various applications of laser processing in development and production of solar cells [3].

**Table 1.** A comparison of laser systems and parameters used for Si micromachining [13].

|  | TRUMPF TruMicro 5000 Series | COHERENT Monaco | IPG PHOTONICS GLPF Series | SPECTRA PHYSICS Spirit Series |
|---|---|---|---|---|
| Operational mode | picosecond, short-pulse, femtosecond | femtosecond | short-pulse | femtosecond |
| Technology | solid-state | solid-state | fiber | solid-state |
| Spectrum | infrared | infrared, green | green | tunable, adjustable wavelength |
| Applications | industrial, for materials processing | industrial, cutting, for materials processing, drilling, pumping, for marking | cutting, drilling, for medical equipment, marking | industrial |
| Wavelength | 1030 nm | 517 nm, 1035 nm | 515 nm | 515 nm, 1,030 nm, 1040 nm |
| Power | Min.: 10 W Max.: 100 W | 20 W, 40 W | 5 W, 10 W, 20 W | Min.: 8 W, Max.: 100 W |

Laser processing of silicon has been investigated with a range of laser systems of various pulse widths and wavelengths. It was observed that longer pulses and wavelengths increase the cutting

speed due to an increased distribution of energy into the silicon. As a result, the process is more prone to thermal effects, such as melting, cracking, amorphization, and residual stress buildup [14]. The use of ultrashort laser pulses has eliminated most of these problems. The ability to process a wide range of materials with negligible thermal effects and without the necessity of post-processing tools, led to innovation in the area of micro and nanotechnology. Although companies like Coherent provide industrial solutions ready to use in production environments, femtosecond lasers are still considered not reliable enough for an efficient commercial use, which requires minimum complexity and maximum stability. Meanwhile, laser pulses with durations in the range of picoseconds are considered very well suited for micromachining of all types of materials and are often a first choice to achieve the desired quality and accuracy at industrial reasonable efforts and costs [15].

The efficiency of the micromachining process depends on both: laser radiation parameters (pulse duration, wavelength, pulse energy, repetition rate) and target material properties (thickness, thermal conductivity, hardness, etc.) [16]. These parameters affect the course of laser ablation. The nature of the ablation process in the femtosecond regime is different and more complex than with longer, i.e., picosecond and nanosecond pulses [17,18]. We found a range between picosecond and femtosecond laser pulses interesting for further investigation, since both short and long pulse types of laser-matter interactions appear. The prototype sub-picosecond laser demonstrated in this paper has already proven to be a sufficient tool for micromachining of thin metal foils and plastics [19] and its application is promising in the photovoltaic industry.

The monocrystalline silicon used in this research is the base material for many types of electronic equipment, integrated circuits, and discrete components. It also serves as a photovoltaic, light-absorbing material in the manufacture of solar cells. According to a report [20], in 2018, laboratory results showed that the best performing modules were based on monocrystalline silicon with 24.4% efficiency. The best results demonstrate the potential for further efficiency enhancement desirable by the mass production level [20].

## 2. Experimental Setup

The micromachining process of the silicon wafers was carried out using a prototype diode-pumped Yb:KYW fiber laser, generating pulses of 650 fs, i.e., in the sub-picosecond regime. Fiber laser technology has such benefits as high average power and temporal stability. In this prototype design, with a working range of 100–400 kHz, the 200 kHz pulse repetition rate proven to be most efficient and ensured temporal stability of the output power. The laser generates a fundamental wavelength of 1030 nm (3 nm FWHM), second (515 nm), and third (343 nm) harmonic. For this experiment, the 343 nm UV beam was used, where the absorption coefficient of a crystalline silicon is at the maximum level [21]. The laser was operating at 5.5 W (27.5 μJ). According to a previous study [22], two ablation regimes are observed for crystalline silicon in the UV range. The ablation thresholds are 0.54 and 2.4 J/cm$^2$. The 5.5 W operating power in our experimental setup translates into laser fluence of 3 J/cm$^2$, which is just above the second ablation threshold. We found that region most suitable to perform micromachining.

From the laser output, the beam was passing through the optical collimator and focusing system, consisting of mirrors and lenses adequate for the applied laser wavelength. After passing through the optical collimator, the laser beam was directed into the positioning system. To move and focus the UV laser beam on the surface of the workpiece, an optical scanner with a focusing lens was applied. The scanner is equipped with two galvanometric mirrors, that deflect the laser beam, making it possible to move within a given pattern in the XY plane. A telecentric lens was used to focus the beam on the sample surface, as it provides a uniform beam interaction with material in the whole scanning area. The samples were put on the z-axis adjustable table, directly beneath the scanning area, on the exact level of the laser beam focus (Figure 2).

The presented setup provided a 2-μm micromachining accuracy and allowed to perform a wide variety of the machining processes: cutting, scribing, engraving, surface structuring, drilling, and dicing.

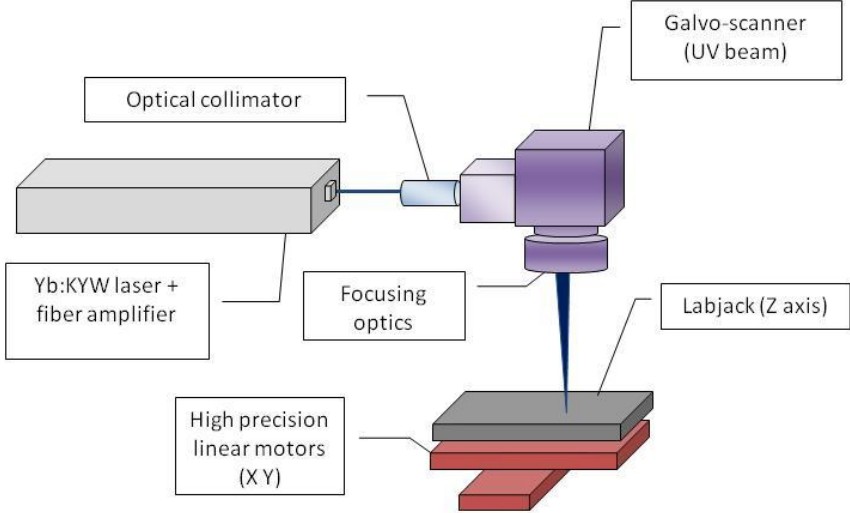

**Figure 2.** The experimental setup.

## 3. Results

In this experiment, 210 µm Si wafers were used as samples. Monocrystalline silicon is known to be very hard but brittle [23], prone to potential damage during the micromachining process. The processing parameters, such as machining speed and number of pattern repetitions, were therefore chosen very carefully and within a limited range. The optimum values were found to be 7 mm/s and between 15 and 20 repetitions (see Figure 3 for reference). There is however a potential to further increase the scanning speed, which will translate to better efficiency, but lower ablation rate. A desired crater depth should be considered according to specific applications.

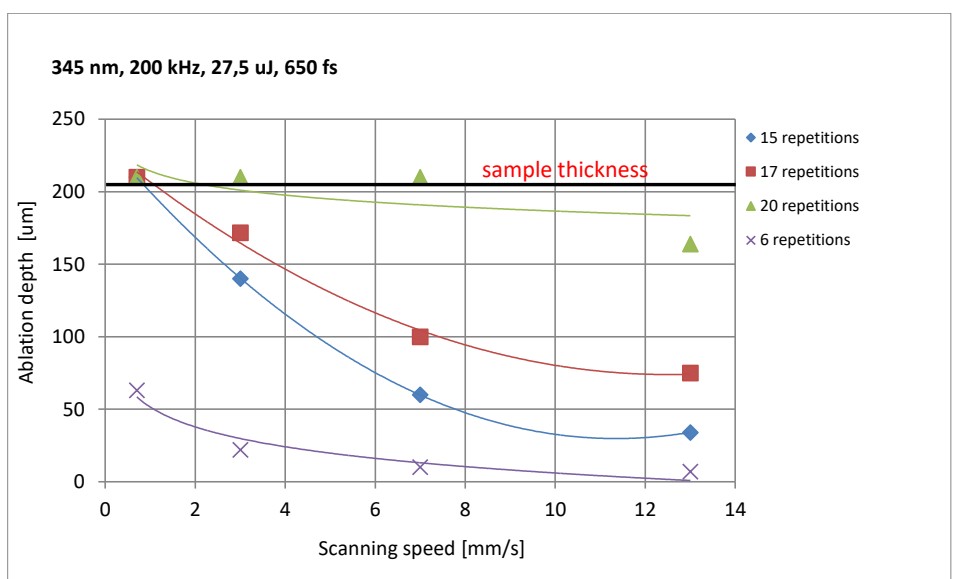

**Figure 3.** The results of a study performed to optimize the machining parameters of 210 µm silicon wafers. The samples were scribed with 4 different values of scanning speed (0.7–13 mm/s), in 4 variants of pattern repetitions (6–15x). The ablation depth of the craters was then measured for each case. Laser beam parameters: 650 fs, 343 nm, 200 kHz, 27.5 µJ.

To analyze the micromachining results, two different approaches were required. The samples, that were cut through during the process, were analyzed by scanning electron microscopy technique (Figure 4). The samples scribed with sub-picosecond laser were analyzed using confocal microscopy.

A laser confocal microscope imaging allowed higher contrast and measurement resolution. The 3D images were created to make crater profile and depth measurements (Figure 5).

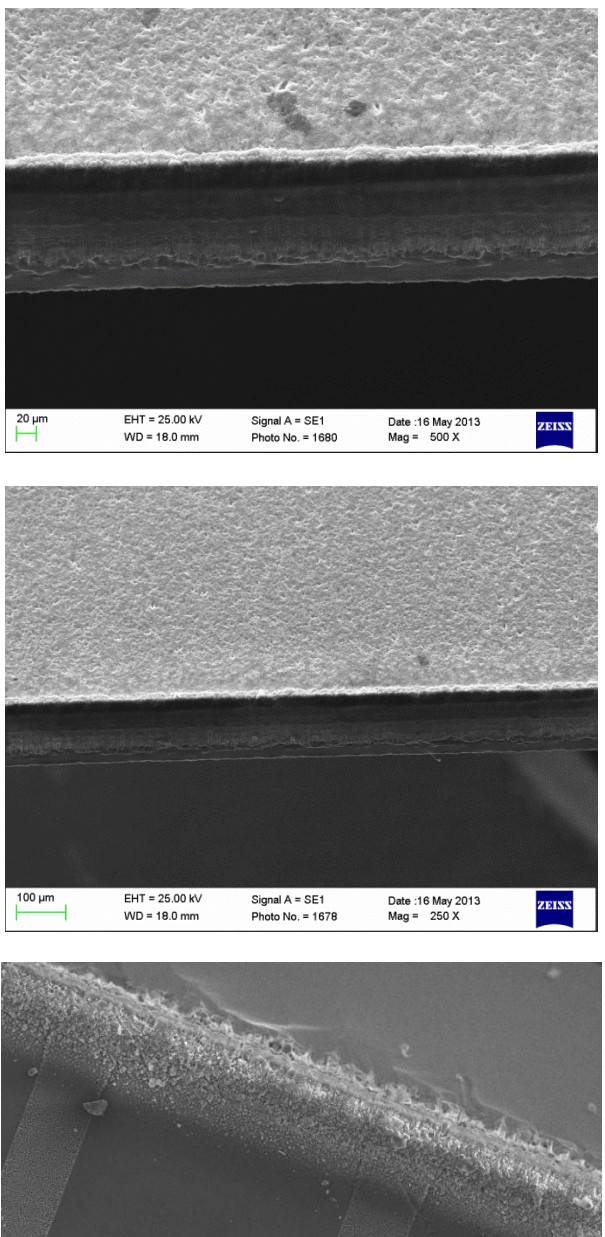

**Figure 4.** SEM images of the Si wafers post sub-picosecond micromachining. Laser beam parameters: 650 fs, 343 nm, 200 kHz, 27.5 µJ; processing parameters: 7 mm/s, 20 pattern repetitions.

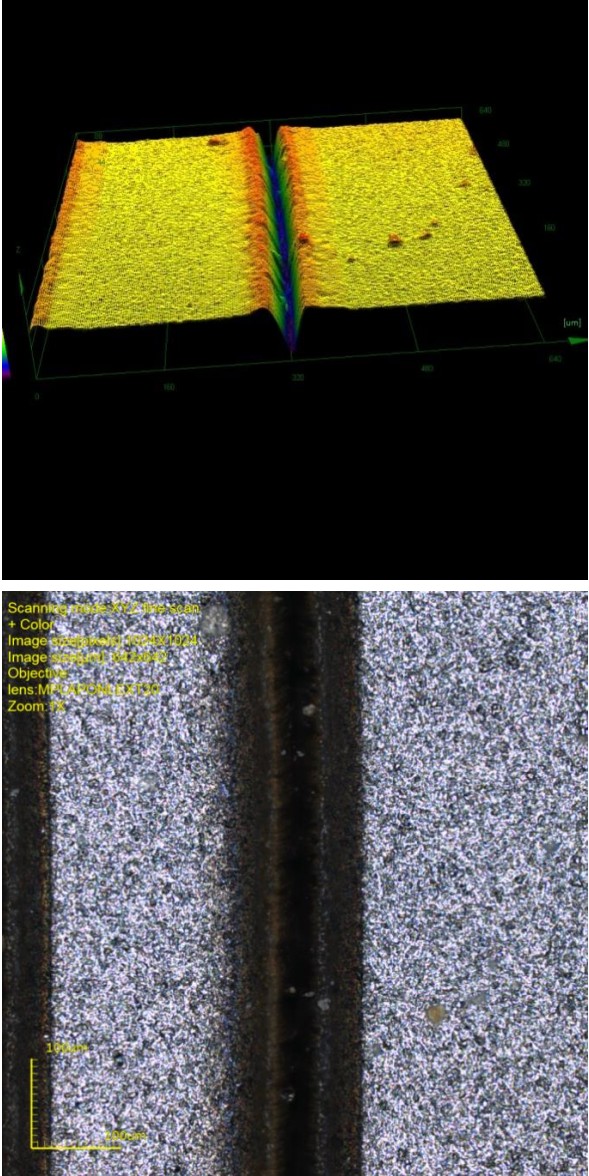

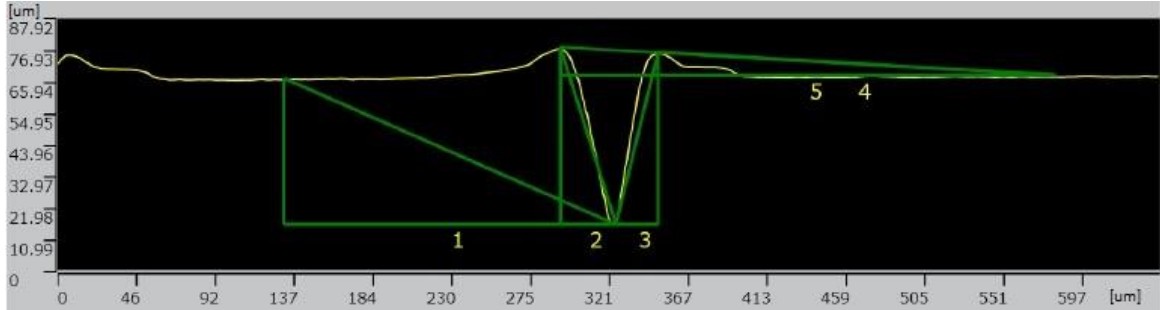

**Figure 5.** Confocal microscopy images and crater profile of the Si wafers post sub-picosecond micromachining. Laser beam parameters: 650 fs, 343 nm, 200 kHz, 27.5 µJ; processing parameters: 7 mm/s, 15 pattern repetitions.

High quality laser micromachining of monocrystalline silicon is difficult to achieve with pulses in the range of pico or nanoseconds [24–26]. The technique is well established for laser pulses of 5 to 400 fs duration [24,27]. Our research shows that applying the pulses of 650 fs, i.e., sub-picosecond range, gave high quality results in both cutting and scribing of the Si wafers.

For ultrashort pulse durations, the fundamental physical processes like energy deposition, melting, and ablation are separated in time. Because the electron-lattice heating and melting occurs roughly on a picosecond timescale [28], the laser-matter interactions in sub-picosecond regime leave the area surrounding the micromachining target unaffected. SEM images show clean surface with lack of micro-cracks or debris. The confocal microscope images indicate a minimal range of a so called heat affected zone. The crater profile shown in Figure 4 has a clear, conical shape, 60-μm depth, and is 55-μm wide (Table 2).

**Table 2.** Numerical results of Si micromachining.

| V [mm/s] | Pattern Repetitions | P [W] | E [μJ] | Wavelength [nm] | Crater Width [μm] | Crater Depth [μm] |
|---|---|---|---|---|---|---|
| 7 mm/s | **15** | 5.5 | 27.5 | 343 | 55 | 60 |
| 7 mm/s | **17** | 5.5 | 27.5 | 343 | 75 | 100 |
| 7 mm/s | **20** | 5.5 | 27.5 | 343 | Cut trough | 210 |

In semiconductors and dielectrics irradiated with high laser intensities, the loss of crystalline order is possible within less than one picosecond. The state of the material after irradiation depends strongly on the type of material and on the laser properties such as intensity and wavelength [29]. The results obtained within our research show that both, the laser beam and processing parameters, are in the optimum range for micromachining of silicon. The fact that the surface surrounding the target area remains unchanged might indicate that the crystalline order of the Si wafers is not affected. However, due to some material changes shown in the confocal microscope images, it should be subjected to further investigation. However, it can be stated that it enables a safe solar cell micromachining, e.g., edge isolation processes.

Figure 6 shows a large scale image of the Si wafers surface hatching that demonstrate the applicability of this process for solar cell micromachining, e.g., for micro-structuration.

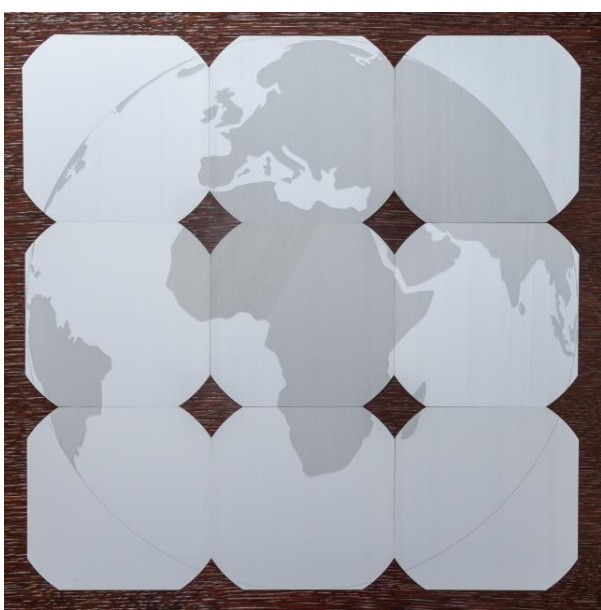

**Figure 6.** *Cont.*

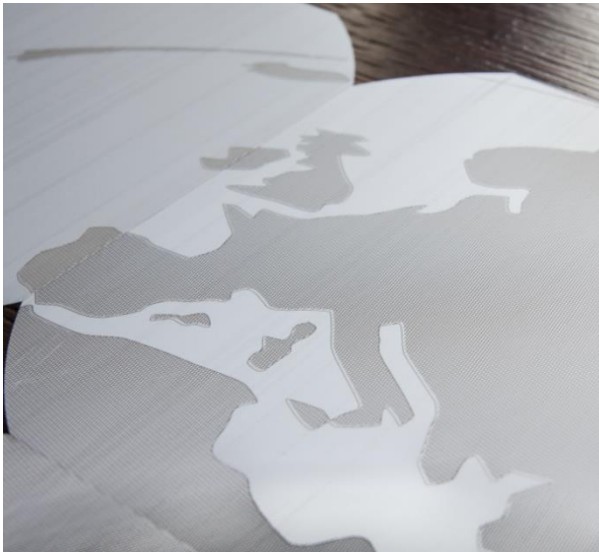

**Figure 6.** Large scale images of the Si wafers sub-picosecond surface hatching. Laser beam parameters: 650 fs, 343 nm, 200 kHz, 27.5 μJ; processing parameters: 7 mm/s, 15 pattern repetitions, total processing time 1.5 h.

## 4. Conclusions

One of the main issues for the photovoltaics scientists and industry is to produce devices with high conversion efficiency, cost-effective, and competitive to traditional methods of generating electricity [30].

With high costs and complexity of femtosecond devices on the one hand, and low efficiency and quality of long pulse micromachining on the other, it is a sub-picosecond laser that might be an alternative when it comes to scribing and cutting of thin silicon wafers. The unwanted thermal effects, i.e., melting, cracking, amorphization, and residual stress buildup, were not observed during the experiment. A sub-picosecond micromachining technique has a potential to be applied in thin film solar cell manufacturing for dicing (cutting, Figure 4), surface structuring (hatching, Figure 6), and edge isolation processes (scribing, Figure 5). The results such as micromachining quality, depth of cut, and crater width, demonstrated during the experiment, can be considered when applying for industrial use.

In a wide variety of micromachining processes, the duration of each one would depend on the desired results and the setup applied (a galvo scanner, linear translation stages, etc.). The micromachining parameters that were found optimum in the presented experiments are considered for the specific applications. There is, however, a potential to further increase the scanning speed or decrease the number of repetitions, which will translate to lower ablation rate, but even better efficiency of the whole process.

Comparing to the commercially available ultrashort pulse lasers described in the introduction (Table 1), we find a sub-picosecond laser more cost efficient and a reliable source. In addition, the prototype Yb:KYW design offers some unique parameters, such as repetition rate (100–400 kHz), UV wavelength, or obtainable laser fluence close to the silicon ablation thresholds. The $M^2$ = 1.2 parameter, indicating a beam quality, is also an advantage for above-mentioned laser.

**Author Contributions:** All authors have seen and approved the manuscript and have contributed significantly to the paper. All authors have read and agreed to the published version of the manuscript.

**Funding:** This research is financed by the Institute of Fluid Flow Machinery, Polish Academy of Sciences O3/Z3/T1.

**Conflicts of Interest:** The authors declare that they have no known competing financial interests or personal relationships that could have appeared to influence the work reported in this paper.

**Data Availability:** The data that support the findings of this study are available from the corresponding author on request.

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
