# Peer review of "Sub-Picosecond Micromachining of Monocrystalline Silicon for Solar Cell Manufacturing"

_applsci, doi:10.3390/app10207277_

Round 1

Reviewer 1 Report

I do not see any major shortcomings in the description of the
studies carried out. The results obtained appear reliable. However, if the discussion of the results and the conclusion reached were not based solely on observation of experiments, the work could be of interest to a wider audience.
In particular, it would be useful to substantiate the advantages
of using sub-picosecond pulses in this particular case of
micromachining, linking the physical parameters of the sample
(Si wafers) to both the laser pulse and the process duration itself.

Author Response

Dear Reviewer,

Thank you very much for your meaningful input. I would like to note, that several sections of the manuscript have been extended according to other reviewers comments. I hope that you will also find it to your satisfaction.

As for linking the physical parameters of the sample to both the laser pulse and the process duration itself, it is quite a demanding task. There is a wide variety of micromachining processes and the duration of each one would depend on the desired results. In some applications (for example edge isolation) where both high processing quality and high ablation depth are needed, the lower machining speed would be preferable. In other, like low depth surface hatching, a high micromachining speed and just a few pattern repetitions are sufficient. This is the reason, why we neglected a comprehensive discussion on this topic, focusing mainly on showing the advantages of sub-picosecond micromachining itself.

Please see the attached manuscript with corrections and comments.

Reviewer 2 Report

The authors applied sub-picosecond laser to process and cut silicon wafers to prove it as a promising tool for solar cell micromachining. Overall, the novelty and significance of the study performed in this article are not clear. The introduction should provide a more updated background and include more details about the different laser systems that have been already used for processing and/or cutting silicon wafers, as well as the parameters applied (machining speed, repetitions) and achieved quality. The results section should include the study performed to optimise the laser parameters to those applied in the experiments (5.5W, 200kHz, 0.7 mm/s, 15-20 repetitions) and a discussion to compare the machining speed, processing time, and quality achieved in the experiments to the current industrial processing/cutting speeds for thin silicon wafers and other laser systems previously used with the same purpose.

Line 19: Add the meaning of PV (photovoltaics).

Line 37: The authors state that femtosecond systems are considered not reliable enough for an efficient commercial use, however, there are already femtosecond industrial lasers able to work for production environments, including applications such as wafer dicing (Monaco femtosecond laser, Coherent, for example).

Figure 3: Quality of SEM images should be improved to show the structure in detail and see if amorphization has occurred.

Figure 4: Add scale (micrometers)

Lines 118-119: I am not sure this can be stated according to the reported results (The fact, that the surface surrounding the target area remains unchanged indicates, that the crystalline order of the Si wafers is not affected). Confocal microscope images show molten material at the edges of the crater and heat affected zone.

Line 121: What is the processing time for the laser-etched silicon wafers shown in Figure 5?

Author Response

Dear Reviewer,

Thank you very much for your meaningful input. Please find our response to the particular comments below.

  1. The introduction should provide a more updated background and include more details about the different laser systems that have been already used for processing and/or cutting silicon wafers, as well as the parameters applied (machining speed, repetitions) and achieved quality.

Thank you for this comment. A comprehensive study of the different laser systems that have been already used for processing and/or cutting silicon is a demanding task. It is due to a wide variety of micromachining processes and different experimental setups used to specific applications. Also, the parameters would differ, depending on the desired results (for example ablation depth or efficiency). In our research we focus mainly on showing the advantage of a sub-picosecond micromachining itself. Therefore, as suggested, we have added some background in the “introduction” section, comparing lasers that could be used for silicon wafers processing. We have also extended the discussion in the “conclusions” section.

  1. The results section should include the study performed to optimize the laser parameters to those applied in the experiments (5.5W, 200kHz, 0.7 mm/s, 15-20 repetitions) and a discussion to compare the machining speed, processing time, and quality achieved in the experiments to the current industrial processing/cutting speeds for thin silicon wafers and other laser systems previously used with the same purpose.

Thank you for this suggestion. We have provided an explanation of the particular choice of the laser pulse parameters in the “experimental setup” section and added a chart with processing parameters optimization results in the “results” section. I hope you will find it to your satisfaction.

Line 19: Add the meaning of PV (photovoltaics).

Explanation added as suggested.

Line 37: The authors state that femtosecond systems are considered not reliable enough for an efficient commercial use, however, there are already femtosecond industrial lasers able to work for production environments, including applications such as wafer dicing (Monaco femtosecond laser, Coherent, for example).

Let us clarify, that it is not the authors intention to suggest, that the femtosecond lasers are not able to work in the production environment. There are however some issues, that make them a less popular choice among the manufacturers, than, for example, picosecond lasers. We have rephrased the paragraph, including your suggestion, and added some reference. We hope that it will provide a better understanding of our previous statement.

Figure 3: Quality of SEM images should be improved to show the structure in detail and see if amorphization has occurred.

Thank you, according to your suggestion, we have added another SEM image to Fig.4.

Figure 4: Add scale (micrometers)

Scale added to figures 5 a-c. Please note, that the figure numeration has increased by 1.

Lines 118-119: I am not sure this can be stated according to the reported results (The fact, that the surface surrounding the target area remains unchanged indicates, that the crystalline order of the Si wafers is not affected). Confocal microscope images show molten material at the edges of the crater and heat affected zone.

Thank you for this meaningful comment. That statement indeed appear a little strong. We have rephrased the paragraph, taking into consideration, that a further investigation might be needed.

Line 121: What is the processing time for the laser-etched silicon wafers shown in Figure 5?

The processing time of all wafers, including wafer switching time and preparation, was approximately 1,5 hr.

Please see the attached manuscript with corrections and comments.

Round 2

Reviewer 2 Report

I would like to suggest just a few additional changes:

1- Abstract: I would add the optimal parameters found in this work to process the silicon wafers: 650fs, UV, 200kHz, 7mm/s, 15 repetitions. And I would add also a brief comment about the advantages of the laser system used in this work in comparison to the commercially available systems (those mentioned by the authors in the conclusions) at the end of the abstract.

2- Figure 3, Caption: Describe a little bit more what is represented in the figure.

3- Figure 6, Lines 146-147: I would add the processing time of the wafers shown in the figure (1.5h).

Author Response

Dear Reviewer,

Thank you for the additional comments. We have completed all the minor revisions as suggested. Please see the manuscript attached.